# Advances in the Preoperative Identification of Uterine Sarcoma

**DOI:** 10.3390/cancers14143517

**Published:** 2022-07-20

**Authors:** Junxiu Liu, Zijie Wang

**Affiliations:** 1Department of Obstetrics and Gynecology, The Affiliated Hospital of Qingdao University, Qingdao 266000, China; 2Department of Urology, The Affiliated Hospital of Qingdao University, Qingdao 266000, China; wzj13256877239@163.com

**Keywords:** uterine sarcomas, preoperative diagnosis, biomarks, radiomics and machine learning, review

## Abstract

**Simple Summary:**

As a lethal malignant tumor, uterine sarcomas lack specific diagnostic criteria due to their similar presentation with uterine fibroids, clinicians are prone to make the wrong diagnosis or adopt incorrect treatment methods, which leads to rapid tumor progression and increased metastatic propensity. In recent years, with the improvement of medical level and awareness of uterine sarcoma, more and more studies have proposed new methods for preoperative differentiation of uterine sarcoma and uterine fibroids. This review outlines the up-to-date knowledge about preoperative differentiation of uterine sarcoma and uterine fibroids, including laboratory tests, imaging examinations, radiomics and machine learning-related methods, preoperative biopsy, integrated model and other relevant emerging technologies, and provides recommendations for future research.

**Abstract:**

Uterine sarcomas are rare malignant tumors of the uterus with a high degree of malignancy. Their clinical manifestations, imaging examination findings, and laboratory test results overlap with those of uterine fibroids. No reliable diagnostic criteria can distinguish uterine sarcomas from other uterine tumors, and the final diagnosis is usually only made after surgery based on histopathological evaluation. Conservative or minimally invasive treatment of patients with uterine sarcomas misdiagnosed preoperatively as uterine fibroids will shorten patient survival. Herein, we will summarize recent advances in the preoperative diagnosis of uterine sarcomas, including epidemiology and clinical manifestations, laboratory tests, imaging examinations, radiomics and machine learning-related methods, preoperative biopsy, integrated model and other relevant emerging technologies.

## 1. Introduction

Uterine sarcomas are a group of relatively rare malignant tumors originating from uterine tissues, with aggressive and poor prognosis [1]. By contrast, hysteromyomas are common benign tumors of the uterus that affect 40–80% of women during their lifetime [2]. Minimally invasive surgery has fewer postoperative complications and enables faster recovery, and laparoscopic comminution of uterine fibroids has become a commonly used minimally invasive surgical procedure. However, for uterine tumors not suspected to be malignant preoperatively or intraoperatively, comminution can lead to the pelvic–abdominal dissemination of the tumor, increasing the rate of disease recurrence and shortening the progression-free and overall survival [3,4]. In December 2017, the U.S. Food and Drug Administration reported that uterine sarcomas affected approximately 1 in 225 to 1 in 580 women who underwent surgery for uterine fibroids [5], while warning of the risk of intraperitoneal morcellation-associated dissemination. Preoperative clarification of the benign and malignant nature of uterine masses can help us select the appropriate surgical approach, avoid medical dissemination, and improve patient prognosis. Herein, the literature on the preoperative diagnosis of uterine sarcomas in recent years has been searched and summarized.

### Methods

The PubMed and Web Of Science database were used as a source. The search terms used were: uterine sarcoma and diagnosis. From the 3480 search results obtained in the period between 2000 and 2022, 213 abstracts were evaluated according to title, 123 articles were read extensively, and 96articles were eventually included. We included population-based studies that included laboratory tests, imaging examinations, radiomics and machine learning-related methods, pre-operative biopsy, integrated model and other relevant emerging technologies. Given the rarity of uterine sarcomas, studies were not limited by design or number of reported patients.

## 2. Epidemiology and Clinical Manifestations

The annual incidence of uterine sarcomas is approximately 1.7 per 100,000 women, with a young peak age of onset, accounting for 3–10% of uterine malignancies [6]. The subtypes of uterine sarcomas are leiomyosarcomas (LMS), endometrial stromal sarcomas (ESS), and adenosarcomas (AS). Carcinosarcomas were classified as dedifferentiated endometrial carcinomas by the International Federation of Gynaecology and Obstetrics in 2009 [7]. There are ethnic differences in the incidence of uterine sarcoma, and the incidence is higher in black women [8], and its risk factors are related to obesity, menopausal use of estrogen and progesterone, oral contraceptives, history of pelvic radiotherapy [9], genetic defects, and use of tamoxifen [10]. The clinical presentation of uterine sarcomas is similar to that of uterine fibroids, which can present with irregular vaginal bleeding, pelvic masses, and pelvic and abdominal pain.

LMS is the most common type of uterine sarcomas, accounting for more than 60% of cases, with an average onset age of 48 years, and the gross specimens are mostly single large masses with soft, fish-like cut surfaces [11], mostly located in the myometrium or subserosal layer of the uterus. Endometrial mesenchymal sarcomas are further subdivided into low-grade endometrial stromal sarcomas (LG-ESS), high-grade endometrial stromal sarcomas (HG-ESS), and undifferentiated endometrial stromal sarcomas (USS). As the second most common type of uterine sarcoma, LG-ESS is less malignant, its onset age has become younger in recent years, and more than 50% of cases occur in premenopausal women. Tumors can be located in the endometrium, myometrium, and subplasma, and the ovary is the most frequent site of extrauterine lesions, usually associated with endometriosis [12]. HG-ESS was reintroduced as a distinct entity in the 2014 World Health Organization classification [13], rare, with a mean onset age of 50 years and biological behavior and prognosis intermediate between LG-ESS and UUS [14]. The tumors often present as intrauterine polyp-like bulges or myometrial nodules. The cut section of tumors shows a fish-flesh appearance, and hemorrhage and necrosis are common. UUS is a highly heterogeneous interstitial uterine tumor with high malignancy, is rare, and usually occurs in postmenopausal women, and pathological diagnosis lacks a specific differential diagnosis and requires diagnosis by exclusion (by extensive sampling) [15]. The tumor forms a tan or white polypoid mass in the uterine cavity with indistinct borders and a fish-like cut surface; necrosis and bleeding are common [16]. Adenosarcoma is a mixed tumor with low malignant potential, consisting of a mixture of benign glandular epithelium and low-grade sarcoma, with a wide age range of onset, occurring mainly in postmenopausal women. The vast majority of AS originate from the endometrium (85%), whereas others originate from the cervix or outside the uterus [17]. The tumors are often polypoid in appearance, with an average diameter of 5 cm.

## 3. Laboratory Tests

Cancer antigen 125 (CA125): Early studies have reported that preoperative serum levels of CA125 were significantly higher in the uterine sarcoma group than in the uterine fibroid group. However, a significant overlap was found between the early stages of uterine sarcomas and uterine fibroids [18]. A subsequent study [19] concluded that preoperative serum levels of CA125 were not predictive of both uterine sarcomas and uterine fibroids. Thus, CA125 is not a good predictor.

Lactate dehydrogenase (LDH): Immunohistochemistry performed on postoperative pathology revealed significantly higher positive rates of LDH-A and LDH-D in uterine sarcomas than in uterine leiomyomas [20]. A Chinese team [21] revealed that LDH ≥ 185 U/L was an independent predictor of LMS. Goto et al. [22] reported that elevated LDH and LDH3 levels combined with contrast-enhanced magnetic resonance imaging (CE-MRI) had high sensitivity and specificity in the diagnosis of LMS. Moreover, Mollo et al. reported that a high LDH5/LDH1 ratio was associated with the development of uterine sarcomas [23]. In 2019, According to an Italian team [24], LDH3 isoforms exhibited better predictive abilities than other isoforms and combining LDH3 with LDH1 isoenzymes into an inverse algebraic relationship, LDH3+ (24/LDH1), named Uterine mass Magna Graecia (U.M.G.) risk index. Uterine sarcomas likely occur when the UMG index is ≥29. In 2020, Lauren et al. [25] validated the U.M.G. index in 179 patients with uterine fibroids, obtained 91.1% specificity, and found that obese women had a higher false-positivity rate than non-obese women. Moreover, LDH combined with MRI [22] and positron-emission tomography-computed tomography (PET-CT) [26,27] could improve the sensitivity and specificity of the preoperative diagnosis of uterine sarcomas.

Neutrophil/lymphocyte ratio (NLR): The inflammatory response is closely related to the tumor, with a high concentration of inflammatory cells in the tumor microenvironment, including elevated neutrophil count and reduced lymphocyte count or function. Several studies have reported that an elevated NLR ratio can be used in the differential diagnosis of uterine sarcomas, with threshold values set at 2.12 [28], 2.1 [29], and 2.8 [21].

Growth differentiation factor-15 (GDF-15): GDF-15 is a potential novel biomarker for the preoperative identification of malignant pelvic diseases [30]. Japanese scholars [31] found that three biomarkers, namely, GDF15, granulocyte precursor protein, and bone bridge protein, could be used for the preoperative differentiation of uterine sarcomas from uterine fibroids through the analysis of the Gene Expression Omnibus and The Cancer Genome Atlas databases. Moreover, they developed a novel measurement system called the compact chemiluminescent immunoautoanalyzer POCube.

MicroRNA: Seven miRNAs were found to be candidate markers for the preoperative identification of uterine sarcomas [32]. When miR-1246 and miR-191-5p were combined, the area under the curve (AUC) of the receiver operating characteristic curve was 0.83 for the diagnostic performance of overall uterine sarcomas and uterine fibroids; specifically, for LMS and uterine leiomyomas, the AUC reached 0.97.

CRP and D-dimer: Patients with LMS had elevated levels of C-reactive protein (CRP) and D-dimer compared with patients with uterine leiomyomas [33]. Combining patients’ preoperative serum LDH, D-dimer, and CRP can be used to identify LMS and uterine leiomyomas, especially degenerative or atypical fibroids [34].

The studied biomarkers in uterine sarcoma in serum was summarized in Table 1.

## 4. Imaging Examinations

### 4.1. Color Doppler Ultrasonography 

Color Doppler ultrasonography is currently the preferred method for the clinical screening of benign and malignant uterine tumors, and it is useful for the initial differentiation of uterine sarcomas from other uterine lesions; thus, uterine sarcomas should be suspected when the tumor diameter is >8 cm [29,35]. However, Chen et al. [36] did not find a significant difference in the tumor size between uterine sarcomas and uterine leiomyomas. Kurjak et al. [37] proposed intra-mass resistance index (RI) ≤ 0.40 as a criterion for the diagnosis of uterine sarcomas, but when a uterine sarcoma has a large liquid-dark area, no blood flow signal is measured inside the tumor, which is easily misdiagnosed as uterine fibroid degeneration, and RI values have been demonstrated to be suboptimal in differentiating benign from malignant tumors [38,39]. A database of 13 ultrasound centers in Italy and Spain [40] analyzed preoperative ultrasound images of 183 patients with uterine sarcomas over 10 years and concluded that the ultrasound characteristics of uterine sarcomas were characterized by large masses with heterogeneous structure, heterogeneous echoes, irregular cystic or necrotic areas, and vascular hyperplasia. Exacoustos [41], Jiyun Oh [42], and others had similar conclusions. American scientists [43] have achieved good performance in predicting mesenchymal malignant lesions in the uterus by applying a radiomics model of ultrasound images. Compared with ultrasonography, MRI has the advantages of good soft-tissue resolution and multidirectional and multiparametric imaging [44], which can more effectively differentiate uterine sarcomas from uterine fibroids; thus, when ultrasonography suggests the possibility of malignant uterine tumors, MRI should be continuously examined.

### 4.2. MRI

MRI can provide a better evaluation of the tumor localization, qualitative diagnosis, and peripheral invasion. LMS can exhibit a high T1 signal [45], but a high T1 signal can also occur when uterine fibroids are associated with hemorrhage. Therefore, a study [46] did not find a significant difference in T1 presentation between uterine sarcomas and uterine fibroids. A high T1 signal within LMS exhibits more heterogeneity, poorly defined boundaries, greater occupancy, and lower signal intensity (SI) than a high T1 signal within uterine fibroids [47]. Compared with uterine leiomyomas, uterine sarcomas have significantly higher SI in T2 [48,49], with central high-signal areas suggestive of necrosis [50,51]. However, some uterine leiomyomas with different degenerations (cystic, mucinous, and effusion) or cellular histological subtypes can also exhibit a high SI on T2-weighted imaging (T2WI) [52], which makes it difficult to determine the benignity or malignancy of uterine tumors. When combining the three measurements of T2 ratio, tumor-myometrial contrast on T2, and tumor-myometrial contrast enhancement ratio, sensitivity can reach 100% without missing any malignant lesions [46]. Namimoto et al. [53] quantified the SI of tumors on T2WI as the tumor-myometrial contrast ratio (TCR) by using the formula. TCR combined with apparent diffusion coefficient (ADC) can completely differentiate sarcomas from uterine fibroids. Several studies [54,55,56,57] have found that diffusion-weighted imaging (DWI) high signal and low ADC values were predictors of uterine sarcoma, whereas slightly different ADC values were determined by other studies. DWI and ADC combined with lymph node enlargement or retroperitoneal masses [58] can improve the detection of uterine sarcomas. The intensity of uterine sarcomas strengthen earlier in CE-MRI [22] and maintain a high level of enhancement after peaking at 60s, whereas uterine leiomyoma strengthens later and to a lesser extent. The diagnostic accuracy [59] and specificity of CE-MRI are significantly higher than those of DWI. In summary, imaging physicians can identify uterine sarcomas by combining multiple parameters: unclear tumor borders, irregular shape, presence of high-signal areas in T1 and/or T2 (caused by hemorrhage and necrosis), high enhancement on CE-MRI, high signal on DWI, and low values of ADC [60,61,62].

In addition to conventional MRI parameters, emerging MRI methods have been used for the preoperative identification of uterine sarcomas. Perfusion-weighted imaging (PWI) determines the degree of blood perfusion in the region of interest based on the distribution of the contrast agent. Malek [63] demonstrated that PWI parameters alone cannot differentiate benign from malignant uterine lesions. However, the analysis of the extracted information in combination with machine-learning methods yields 91.7% accuracy, 100% sensitivity, and 90% specificity. Magnetic resonance spectroscopy (MRS) can provide markers of biochemical processes associated with metabolic information, transformation of normal to malignant tissues, and presence of active tumors [64]. MRS has been used to examine characteristic metabolite peaks in various diseases. The proportion of choline and lipid peaks in malignant uterine lesions is significantly higher than that in benign lesions [65], and the accuracy was 98.3% in differentiating uterine sarcomas from uterine fibroids by combining ADC and MRS. Another study [66] confirmed that the choline and lipid-positive peaks of MRS were highly suggestive of uterine malignancies. Susceptibility-weighted MR sequences (SWS) take advantage of inter-tissue magnetic susceptibility differences and blood oxygen level-dependent effects to form images. Takeuchi [67] claimed that signal voids on SWS were observed in all sarcoma tissues, but only 4% in uterine fibroids, and SWS had 97% accuracy, 100% sensitivity, and 96% specificity in differentiating uterine sarcomas from uterine leiomyomas. Enhanced T2 star-weighted angiography (ESWAN), as a complement to SWS, is an emerging MRI contrasted imaging technique in recent years, which can obtain not only information such as phase and amplitude, but also quantitative parameters such as phase value, T2* value, and R2* value through high-resolution, thin-layer three-dimensional scanning based on the difference in magnetic-sensitive signals between tissues [68]. Two imaging physicians measured the parameters of ESWAN sequences in two groups of patients [69]. They found that the phase and T2* values in the uterine sarcoma group were greater than those in the benign group, whereas R2* values were less than those in the benign group, which had a differential diagnostic significance. MR diffusion kurtosis imaging (DKI) investigates the diffusion properties of water molecules based on a non-Gaussian distribution model. Two imaging physicians [70] have analyzed DKI of 13 patients with uterine sarcomas and 26 patients with uterine leiomyomas, and the mean kurtosis, axial kurtosis, and radial kurtosis values of the uterine sarcoma group were greater than those of the uterine fibroid group. The mean diffusivity, axial diffusivity, radial diffusivity, and fractional anisotropy values were smaller than those of the uterine fibroid group; all these parameters have high diagnostic performance, but studies involving emerging MRI techniques are scarce and need further validation. The related studies of MRI were summarized in Table 2. 

### 4.3. CT

The role of CT in the preoperative evaluation of suspicious uterine masses is often overlooked. A case report [23] stated that a patient with a postoperative diagnosis of uterine sarcomas was not diagnosed with malignancy by ultrasonography, MRI, and LDH, whereas an elevated LDH5/LDH1 ratio and CT suggested sarcoma. Yu Lan et al. [71] also suggested the importance of CT in ESS, where most ESS lesions are located in the uterine cavity and tumors grow outward, disrupting the uterine cavity and forming outward ruptures or channels, which may appear as hypointense on CT and non-enhancing on enhancement scans. Moreover, the solid areas of the tumors may show significant enhancement in the arterial phase and persistent inhomogeneous enhancement in the venous and delayed phases.

### 4.4. PET-CT

A meta-analysis [72] revealed that PET-CT can distinguish uterine leiomyomas and uterine sarcomas well. When the maximum standard uptake value (SUVmax) was used as a threshold of 7.5 [27], the sensitivity and specificity were 80.8% and 100%, respectively, which could exclude most uterine fibroids, and the negative predictive value (NPV) was up to 100% when SUVmax of 4.4 was used as a threshold. Texture analysis of PET-CT images [73] yielded a diagnostic performance of 100% sensitivity, 94% specificity, and 95% accuracy when combining traditional features and SUV. The 3′-Deoxy-3′-18F-fluorothymidine (18F-FLT) PET-CT allows the evaluation of inflamed tissues for tumor proliferation without the uptake of tracers and has been used in differentiating uterine sarcomas from uterine leiomyomas. It was superior to the conventional PET-CT imaging agent 18F-FDG [74]; however, it is not yet widely used in clinical practice.

### 4.5. Machine Learning and Radiomics

As mentioned previously, MRI can be used to identify uterine sarcomas and uterine leiomyomas, but MRI carries a certain risk of misdiagnosis because uterine leiomyomas often show degeneration and abnormalities in growth patterns, which are difficult to distinguish from uterine sarcomas with the naked eye. By extracting features from medical images that are difficult to discern with the human eye in high-throughput extraction, filtering, downscaling, and modeling of the features, radiomics can help imaging physicians make rapid diagnoses and provide decision support. With the development of computer technology, radiomics based on machine and deep learning are gradually applied to various fields of medical diagnosis, disease identification, and prognosis assessment. Nakagawa et al. [75] significantly improved the preoperative prediction of uterine tumors using machine-learning algorithms that integrated some features from MRI and PET-CT. Lakhman et al. [61] demonstrated that radiomics based on T2WI were feasible in distinguishing LMS from atypical uterine leiomyomas. For uterine tumors with high T2 signal, radiomics combined with clinical variable models [76,77] can achieve predictive performance higher than that of imaging physicians. Malek et al. [57] obtained the best simple decision tree with 96.2% accuracy, 100% sensitivity, and 95% specificity by analyzing 13 features of multiparametric MRI with a machine-learning algorithm, whereas the accuacy, sensitivity, and specificity of the complex tree were up to 100%. Good progress has also been made in radiomics based on ADC maps to differentiate uterine sarcomas from uterine leiomyomas [78,79]. The related studies of machine learning, texture analysis, and radiomics were summarized in Table 3.

## 5. Preoperative Biopsy and Intraoperative Freezing

Pathology is the gold standard for the diagnosis of uterine sarcoma. Preoperative pathology of uterine sarcomas can be obtained through diagnostic curettage, hysteroscopic endometrial biopsy, and ultrasound-guided puncture biopsy. While LMS lesions are mostly located in the myometrium, ESS and AS can present as polypoid masses in the uterine cavity; therefore, the positive rate of diagnostic scraping for LMS is only 42.9%, whereas ESS can reach 83.3% [80]. Hysteroscopy allows the visualization of uterine cavity masses and should alert the possibility of malignancy when the masses have bleeding, necrosis, and irregular morphology. Biopsy under hysteroscopic view also greatly improves the positive rate of endometrial biopsy, but it still cannot reach the muscular layer to obtain the specimen. Biopsy of lesions using fine-needle aspiration can be very useful in distinguishing uterine sarcomas and uterine fibroids [81,82]. Alboni et al. [83] used transabdominal or transvaginal access aspiration biopsy in 10 patients with imaging suggestive of malignancy to palpate deep subplasma or myometrial tumors and performed 4–5 biopsies for each tumor to reduce sampling error. In all patients who underwent surgery, no signs of seeding of biopsy debris or intrapelvic peritoneal spread were visible to the naked eye. Peters et al. [84] performed a 3-year follow-up of patients who underwent preoperative puncture biopsy, and we can expect the possibility of a pre-operative pathological diagnosis of atypical uterine muscle tumors by vaginal ultrasound-guided biopsy. With the development of radiomics, it was proposed [85] to preprocess uterine tissue biopsy images of patients at risk of LG-ESS using segmentation and staining normalization algorithms, applying various classical machine learning and advanced deep learning models to classify tissue images as benign or malignant, with an AUC of 0.87 for the optimal model, suggesting that properly trained learning algorithms can help in pathology reading. For uterine tumors that are highly suspected malignant masses in operation, rapid frozen section pathology testing can help diagnose and determine the extent of surgery.

## 6. Integrated Model

Disease identification and diagnosis are complex, and clinical diagnosis of the disease is mostly made by combining various parameters, such as patient’s age, clinical manifestations, laboratory tests, and imaging examinations. Nagai et al. proposed a preoperative diagnostic scoring system for uterine sarcomas consisting of four parameters, namely, age, serum LDH level, MRI findings, and endometrial cytology examination in 2014 [86] and revised it 1 year later, which improved the diagnostic accuracy [87]. In 2019, Koehler et al. [88] proposed a preoperative scoring model (pLMS) for LMS based on features such as abnormal uterine bleeding, menorrhagia, dysmenorrhea, suspicious ultrasound presentation, and tumor diameter, and a score >1 confirms the diagnosis of uterine sarcomas and a score between −3 and 1 suggests additional investigations. However, Condic et al. [89] validated it using the data of their patients and concluded that the pLMS score was not a reliable tool for predicting LMS, and they did not recommend it for clinical application. Risk prediction models include age, race, body mass index, number of myomas, uterine size, degree of uterine enlargement, and level of pelvic pain, which are equally unsatisfactory [90,91].

## 7. Molecular Genetic Imaging Techniques

Molecular genetic imaging is evolving from a valuable preclinical tool to a reality that drives the clinical management of patients. The technology involves pairing an imaging reporter gene with a complementary imaging agent in a system that can be used to measure gene expression or protein interactions or to track genetically tagged cells in vivo. American scientists demonstrated the ability of survivin promoter-based genetic imaging to accurately differentiate sarcomas from benign tumors in human cells and mouse models of LMS [92], and it may be a candidate for future development and optimization.

## 8. Discussion and Conclusions

Despite their relative rarity, uterine sarcomas are aggressive and have diverse pathological types, atypical clinical symptoms, and severely poor prognosis. Therefore, early identification is critical in improving patient prognosis. Limited by the level of primary care hospitals, successful preoperative prediction of uterine sarcomas and referral of patients to an experienced hospital will significantly improve patient survival [93].

Currently, there are advantages and disadvantages to each of the clinical methods used to identify uterine sarcoma preoperatively. Elevated serum markers CA-125, LDH, CRP, and D-dimer may suggest uterine sarcoma, but are susceptible to other factors and lack specificity. Ultrasound is cheap and convenient, and is the best screening method, but does not well identify the benignity or malignancy of uterine masses. Although MRI has good soft tissue resolution, certain degenerative types of uterine fibroids have similar signal intensities and there is a certain rate of misdiagnosis. PET-CT has the highest accuracy but is expensive and difficult to promote. Preoperative biopsy pathology is the gold standard for diagnosis, but requires a high level of physician skill and has the potential for medical seeding and inadequate sampling for missed examinations. Clinical manifestations such as rapidly growing tumors and elevated serum markers in patients, abnormalities in imaging examinations can alert clinicians to avoid crushing masses of unknown nature.

Given the increasing knowledge on uterine sarcoma, more and more preoperative diagnostic methods have been proposed, but we still face many challenges. First, most studies are single-center studies that suffer from insufficient data size, lack of external validation, and poor model robustness. Second, most studies are based on retrospective data analyses, lack standardized protocols, and have insufficient follow-up time. Finally, with increasing multidisciplinary integration, gynecologists should dabble in emerging fields such as artificial intelligence and molecular biology in addition to their superb clinical skills, which poses a higher challenge to clinicians [94]. Pergialiotis [95] reported that these problems will be gradually solved with broader inter-institutional and international collaboration. We can facilitate research by creating a gynecologic cancer database and recruiting talent in computer science and basic medicine.

Overall, given the surprisingly high prevalence of unsuspected uterine sarcomas and the worsened outcomes associated with laparoscopic power morcellator–assisted hysterectomy or myomectomy, there is increasing pressure to identify uterine sarcoma preoperatively. It will take some time for new tumor marker tests to be implemented in medical institutions. Recent findings relying on multiparametric imaging and imaging histology of MRI to differentiate uterine sarcomas from uterine fibroids are encouraging but still challenging. Long-term results of preoperative puncture biopsy are also under observaion. We should keep passionate about studying uterine sarcomas in the future.

## Figures and Tables

**Table 1 cancers-14-03517-t001:** Overview of studied protein biomarkers in uterine sarcoma in serum or plasma.

Marker	Author	Year	Tumor Type (N)	Controls (N)	Results/Conclusions
CA125	Juang [18]	2006	LMS(42)	UM (84)	Preoperative serum CA125 had a potential role in the differential diagnosis between early stage and advanced-stage uterine leiomyosarcoma.
	Yilmaz [19]	2009	USM * (26);	UM (2382)	In the differential diagnosis of myoma and uterine sarcoma, the preoperative serum CA 125 level did not have any predictivity.
LDH	Song [20]	2018	USM (50)	UM (26)	The positivity rates for LDH-A and LDH-D were significantly higher in patients with uterine sarcoma compared with those with uterine myoma.
	Zhang [21]	2020	LMS (45)	UM (180)	LDH ≥ 193 U/L was independent predictors of LMS.
	Goto [22]	2002	LMS (10)	DLM (130)	The combined use of dynamic MRI and serum measurement of LDH seems to be useful in making a differentiated diagnosis of LMS from DLM.
	Di Cello [24]	2019	USM (43)	UM (2211)	UMG, the accuracy of markers in discriminating between benign and suspicious malignant uterine masses was significantly enhanced, sensitivity at 100% and specificity at 99.6%.
	Spivack [25]	2021	Null	UM (179)	Specificity of UMG index to exclude uterine sarcoma was 91.1% (163/179) and higher in non-obese (BMI < 30; 95.1%) than obese women (85.5%).
	Nagamatsu [26]	2010	USM * (10)	UM (24)	The diagnostic accuracy of FDG-PET combined with serum LDH was 100%.
	Kusunoki [27]	2017	USC (15)	UM (19)	PET/CT and LDH levels had a sensitivity of 86.6%, specificity of 100%, positive predictive value of 100%, and an NPV of 90.4%.
NLR	Zhang [21]	2020	LMS (45)	UM (180)	NLR ≥ 2.8 were independent predictors of LMS.
	Kim [28]	2010	USM * (55)	UM (330)	NLR was more powerful for the preoperative diagnosis of uterine sarcomas than serum CA-125 levels.
	Cho [29]	2016	USM(31)	UM (93)	NLR > 2.1 was independent risk factors for uterine sarcoma.
GDF-15	Trovik [30]	2014	USM (19)	UM (50)	The median circulating GDF-15 concentration was elevated in the uterine sarcoma group (943 ng/L) compared with the myoma uteri group (647 ng/L).
MicroRNA	Yokoi [32]	2019	USM (10)	UM (18)	The optimal model consisted of two miRNAs (miR-1246 and miR-191-5p), with an area under the receiver operating characteristic curve (AUC) for identifying LMS of 0.97.
CRP+D-dimer	Nishigaya [34]	2019	USM (36)	UM (97)	When LDH, D-dimer, and CRP were all positive, specificity and positive predictive value were 100% in differentiating leiomyosarcoma from uterine myoma.

LMS: leiomyosarcoma; UM: uterine myoma; USM: uterine sarcoma; DLM: degenerated leiomyoma; CA125: Cancer antigen 125; LDH: Lactate dehydrogenase; NLR: Neutrophil-to-lymphocyte ratio; CRP: C-reactive protein; GDF-15: Growth differentiation factor-15; *: including carcinosarcoma.

**Table 2 cancers-14-03517-t002:** Summary of MRI studied.

MRI	Author	Year	Tumor Type (N)	Controls (N)	Results/Conclusions
T1	Tanaka [45]	2004	LMS (9)/ SMTUMP (3)	UM (12)	It was found that 9 of the 12 nonbenign characters had more than 50% of high-intensity areas on T2-weighted images (T2WI), and some hyperintense foci on T1-weighted images (T1WI).
	Malek [46]	2019	USM (21)	UM (84)	Intensity at T1-weighted sequences exhibited no significant difference between USM and UM (*p* = 0.201).
	Ando [47]	2018	LMS (14)	LM (1118)	T1 HIA within LM showed more homogeneity, better demarcation, smaller occupying rate, and higher signal intensity than T1 HIA within LMS.
T2	Sahdev [48]	2001	USM (22)	\	On T2WI, the masses were characteristically of low or intermediate background signal intensity with pockets of very high T2 signal. The areas of high T2 signal corresponded to cystic necrosis in the tumor.
	Kim [49]	2018	ESS (18) LMS (15)	LCD (30)	ESS or LMS more frequently showed high T2 SI compared with LCD (OR = 4.396; *p* = 0.046).
	Malek [46]	2019	USM (21)	UM (84)	T2-scaled ratio, tumor myometrium contrast ratio on T2 and tumor myometrium contrast-enhanced ratio achieved a sensitivity of 100% in predicting USM.
	Namimoto [53]	2009	USM (8)	UM (95)	A combination of ADC and TCR achieved a significant improvement without any overlap between sarcomas and leiomyomas (sensitivity 100%, specificity 100%).
DWI&ADC	Thomassin-Naggara [54]	2013	USM (25)	UM (26)	The significant criteria for prediction of malignancy were high DWI signal intensity (OR = +∞), intermediate T2-weighted signal intensity (OR = +∞), mean ADC (OR = 25.1).
	Li [55]	2017	LMS (16)	DLM (26)	The mean ADC value in LMS was significantly lower than that in DLMs (*p* < 0.001).
	Sato [56]	2014	LMS (10)	UM (83)	The LMS were readily apparent via DWI, presenting as an intermediate- to high-intensity area in the uterine wall. All low-intensity areas presented as leiomyoma nodules.
	Wahab [58]	2020	USM (51)	UM (105)	Predictive MRI criteria for malignancy were enlarged lymph nodes or peritoneal implants, high DWI signal greater than that in the endometrium, and ADC less than or equal to 0.905 × 10^−3^ mm^2^/s.
CE-MRI	Goto [22]	2002	LMS (10)	DLM (130)	The contrast enhancement at 60s after administration of Gd-DTPA was detected in all LMS but absent in 28 of 32 DLM patients.
	Lin [59]	2016	LMS/SMTUMP (8)	UM (25)	For prospective differentiation between uterine LMS/STUMP and benign leiomyoma, CE-MRI can provide accurate information and is preferable to DWI. A combination of DWI and ADC values can achieve comparable diagnostic accuracy to CE-MRI.
Multi-MRI	Lakhman [61]	2017	LMS (10)	ALM (14)	Four qualitative MR features most strongly associated with LMS were nodular borders, hemorrhage, “T2 dark” area(s), and central unenhanced area(s).
PWI	Malek [63]	2019	USM (10)	UM (50)	When 21 features extracted from ROIs were fed into the classifier an accuracy of 91.7%, sensitivity of 100%, and specificity of 90% were achieved in the optimal operating point of the classifier.
MRS	Rahimifar [65]	2019	USM (21)	UM (84)	The percentage of malignant lesions for which choline and lipid peaks were present was significantly higher than that of benign lesions. By combining the ADC and MRS findings, an accuracy of 98.3 (95.1–100) was achieved.
	Takeuchi [66]	2013	USM (12)	UM (26)	The presence of a high lipid peak for the diagnosis of sarcoma had a sensitivity of 100%, specificity of 96%, positive predictive value of 92% and negative predictive value of 100%.
SWS	Takeuchi [67]	2019	USM (10)	UM (24)	The accuracy, sensitivity, and specificity for SWS were 97%, 100%, and 96%, respectively.
ESWAN	Tian [69]	2020	USM (17)	DLM (33)	The AUC values of phase, R2* and T2* in USM group were 0.854, 0.900 and 0.961, respectively.
DKI	Ju [70]	2021	USM (13)	DLM (26)	The AUC values of MK, Ka, Kr, FA, MD, Da and Dr were 0.93, 0.99, 0.80, 0.73, 0.94, 0.97 and 0.90. The diagnostic threshold of the parameters were as follows: MK ≥ 0.80, Ka ≥ 0.73, Kr ≥ 0.75, FA ≥ 0.22, MD ≤ 1.47, Da ≤ 1.95, Dr ≤ 1.23.

LMS: leiomyosarcoma; UM: uterine myoma; USM: uterine sarcoma; DLM: degenerated leiomyoma; SMTUMP: smooth muscle tumors of uncertain malignant potential; ESS: endometrial stromal sarcomas; LCD: Leiomyoma with cystic degeneration; ALM: atypical leiomyoma; T1 HIA: Hyperintense area on T1 weighted images.

**Table 3 cancers-14-03517-t003:** Summary of machine learning, texture analysis, and radiomics studied.

Methods	Author	Year	Tumor Type(N)	Controls (N)	Results/Conclusions
machine learning	Nakagawa [75]	2018	USM (11)	UM (56)	The AUCs of the univariate models using MRI parameters (0.68-0.8) were inferior to that of the maximum standardized uptake value (SUVmax) of PET (0.85); however, the AUC of the multivariate LR model (0.92) was superior to that of SUVmax, and comparable to that of the board-certified radiologists (0.97 and 0.89).
	Malek [57]	2020	USM (21)	UM (84)	The simple decision tree and a complex one were proposed using the most accurate models. Our final simple decision tree obtained accuracy = 96.2%, sensitivity = 100% and specificity = 95%, while the complex tree yielded accuracy, sensitivity and specificity of 100%.
Radiomics	Lakhman [61]	2017	LMS (10)	ALM (14)	Sixteen texture features differed significantly between LMS and ALM (*p*-values: < 0.001–0.036). Unsupervised clustering achieved accuracy of 0.75 (sensitivity: 0.70; specificity: 0.79).
	Nakagawa [76]	2019	USM (30)	UM (50)	The AUC for the eXtreme Gradient Boosting was significantly higher than those for both radiologists (0.93 vs. 0.80 and 0.68, *p* = 0.03 and *p* < 0.001, respectively) in the differentiation of uterine sarcomas from leiomyomas with high signal intensity on T2WI.
	Wang [77]	2021	USM (53)	UM (81)	Comparing with the T2WI-based radiomics model (AUC: 0.76 ± 0.09) and the clinical model (AUC: 0.79 ± 0.09), the combined model significantly improved the AUC value to 0.91 ± 0.05 (*p* < 0.05). The clinical-radiomics combined model yielded equivalent or higher performance than two radiologists (AUC: 0.78 vs. 0.91, *p* = 0.03; 0.90 vs.0.91, *p* = 0.13).
	Xie [78]	2019	USM (29)	UM (49)	Diagnosis efficacy of radiologists based on MRI reached an AUC of 0.752, sensitivity of 58.6%, specificity of 91.8%, and accuracy of 79.5%. The optimal radiomic model reached an AUC of 0.830, sensitivity of 76.0%, average specificity of 73.2%, and accuracy of 73.9%.
Texture analysis	Niu [79]	2019	USM (16)	DUF (31)	The maximum, mean, standard deviation, 50th, 75th, 90th, 95th, skewness and entropy of USM were less than DUF. The energy value and consistency were greater than DUF, and the differences were statistically significant (*p* < 0.05). The area under the curve (AUC) of the entropy value is the largest, and the diagnostic efficiency is the best (AUC = 0.94).

LMS: leiomyosarcoma; UM: uterine myoma; USM: uterine sarcoma; ALM: atypical leiomyoma; DUF: degenerative uterine fibroids.

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
