# Peer review of "Advances in the Preoperative Identification of Uterine Sarcoma"

_cancers, 2022, doi:10.3390/cancers14143517_

Round 1
Reviewer 1 Report
Dear authors,
some ideeas regarding the manuscript
pg 1 line 42 comminution .....i think intraperitoneal morcellation is more accurate
pg 2 line62 suplasma layer of the uterus ... is that subserosal layer that you wanted to write?
pg 2 line 72 the cut surface is fichy .... please rephrase
pg4 line 103 uterine Magna Graecia index should be developed and presented in a few words
Discussion chapter is too slim, and shoul be consistently developed
it should include what the authors considerations regarding the strengths and weaknesses of the methods that are today available in order to identify prior surgery patients with high risk for sarcoma
Reviewer 2 Report
Nice review, however i believe a short paragraph mentioning the electronic search in a systematic review approach would increase the strength of the paper. Also, some grammatical mistakes are met, the references should have the type of letters with the main body. In addition the text should be double-spaced.
Round 2
Reviewer 1 Report
Congratulations for your work